# Host Derivation of Sindbis Virus Influences Mammalian Type I Interferon Response to Infection

**DOI:** 10.3390/v15081685

**Published:** 2023-08-03

**Authors:** John M. Crawford, Aaron M. Buechlein, Davis A. Moline, Douglas B. Rusch, Richard W. Hardy

**Affiliations:** 1Department of Biology, Indiana University, Bloomington, IN 47405, USA; johmcraw@iu.edu (J.M.C.); damoline@iu.edu (D.A.M.); 2Center for Genomics and Bioinformatics, Indiana University, Bloomington, IN 47405, USA; abuechle@indiana.edu (A.M.B.); drusch@indiana.edu (D.B.R.)

**Keywords:** arbovirus, alphavirus, type I IFN, translational shut-off

## Abstract

Arboviruses are defined by their ability to replicate in both mosquito vectors and mammalian hosts. There is good evidence that arboviruses “prime” their progeny for infection of the next host, such as via differential glycosylation of their outer glycoproteins or packaging of host ribosomal subunits. We and others have previously shown that mosquito-derived viruses more efficiently infect mammalian cells than mammalian-derived viruses. These observations are consistent with arboviruses acquiring host-specific adaptations, and we hypothesized that a virus derived from either the mammalian host or mosquito vector elicits different responses when infecting the mammalian host. Here, we perform an RNA-sequencing analysis of the transcriptional response of Human Embryonic Kidney 293 (HEK-293) cells to infection with either mosquito (Aedes albopictus, C7/10)- or mammalian (Baby Hamster Kidney, BHK-21)-derived Sindbis virus (SINV). We show that the C7/10-derived virus infection leads to a more robust transcriptional response in HEK-293s compared to infection with the BHK-derived virus. Surprisingly, despite more efficient infection, we found an increase in interferon-β (IFN-β) and interferon-stimulated gene (ISG) transcripts in response to the C7/10-derived virus infection versus the BHK-derived virus infection. However, translation of interferon-stimulated genes was lower in HEK-293s infected with the C7/10-derived virus, starkly contrasting with the transcriptional response. This inhibition of ISG translation is reflective of a more rapid overall shut-off of host cell translation following infection with the C7/10-derived virus. Finally, we show that the C7/10-derived virus infection of HEK-293 cells leads to elevated levels of phosphorylated eukaryotic translation elongation factor-2 (eEF2), identifying a potential mechanism leading to the more rapid shut-off of host translation. We postulate that the rapid shut-off of host translation in mammalian cells infected with the mosquito-derived virus acts to counter the IFN-β-stimulated transcriptional response.

## 1. Introduction

Alphaviruses are arthropod borne viruses (arboviruses) predominantly vectored between vertebrates by mosquitoes that are split into two main groups based on the disease they cause. The first group is the arthritogenic viruses, such as the Sindbis virus (SINV) and the chikungunya virus (CHIKV) that cause debilitating but non-life-threatening polyarthritis. The second group is the encephalitogenic viruses, such as the Venezuelan and the Eastern equine encephalitis virus (VEEV and EEEV, respectively) that can cause fatal encephalitis. There are currently no effective vaccines for alphaviruses and outbreaks, particularly of CHIKV, occur regularly around the world. Therefore, in order to develop effective transmission intervention strategies, it is important to improve our understanding of alphavirus replication and interaction with both vertebrate and insect hosts.

SINV is a member of the Togaviridae family and is the type species of the Alphavirus genus. SINV is an enveloped, single-stranded, positive-sense RNA virus with a genome of 11.7 kb. In vertebrate cells, SINV causes an acute, cytolytic infection, whereas in arthropod cells the infection is persistent and non-cytolytic. The enzootic alphavirus transmission cycle consists of a series of stepwise events that alternate between the hematophagous mosquito vector and the vertebrate host. For vertebrate infection, an arbovirus-infected mosquito vector bites a naïve vertebrate while blood feeding. The introduction of virus into the vertebrate host is accompanied by mosquito saliva which contains various proteins with angiogenic, antihemostatic, anti-inflammatory, and immunomodulatory properties, some of which may play a role in facilitating viral transmission [1,2,3]. In the preliminary stages of infection in a vertebrate, the virus encounters multiple cell types, including epithelial cells, dendritic cells (DCs), and various immune cells. While the initial site of infection is not clearly defined for each alphavirus, there is evidence of CHIKV infecting dermal fibroblasts initially while for VEEV it appears to be dermal DCs, Langerhans cells, or keratinocytes [4,5,6,7]. Regardless of the site of initial infection, the infection and subsequent replication in dermal immune cells appears to be an important early step for most arboviruses in establishment and dissemination of infection [4,6,7,8]. Once the virus establishes infection in these dermal immune cells, dissemination is achieved through viral release in the blood stream, as well as drainage of infected plasmacytoid DCs to the lymph nodes which grants access to the entirety of the vertebrate host, where the virus goes on to infect various tissues, muscles, and organs [4,6,7,9]. For arthritogenic viruses, cells in the joints are infected, and disease manifests through active replication of the virus in the area driving the expression of proinflammatory chemokines and cytokines leaving the joint in a constant state of inflammation [4,10,11,12].

The viral transmission cycle mandates that arboviruses are adept at interacting with two very distinct hosts. Currently, there is no evidence to suggest a selection and reselection process occurring during transmission between insect and vertebrate hosts. However, we and others have published work indicating that the viruses produced in insect cells do differ from those produced in vertebrate cells, albeit not at the genetic level [13,14,15,16,17,18]. There is good evidence of arbovirus capabilities to prime the progeny produced in arthropod cells for infection in vertebrate cells as it has been shown that arthropod-derived alphaviruses have increased infectivity on vertebrate cells, and vice versa [13]. Similarly, it has been shown that alphaviruses acquire host-specific alterations that improve their ability to establish infection in the new host; these host-specific alterations range from differential packaging of host components such as 18s rRNAs that improve the viral kinetics in mammalian cells to specific glycosylation of the virus that allows the targeted infection of certain mammalian cell types [13,14,15,16,17]. Also, though it is a finding of a newer field of study, there is evidence of specific RNA modifications of viral RNA in the host that have been shown to have various effects on viral replication. For instance, it has been shown that N6-methyladenosine (m6A) post-transcriptionally regulates the RNA function of the flavivirus HCV in a negative manner in mammalian cells [19]. There is also evidence of another RNA modification, 5-methycytosine (m5C), that SINV acquires in arthropod cells and that has been shown to be pro-viral for infection back onto mammalian cells [20]. Finally, our previous study showed that mosquito- and mammalian-derived SINV have significantly different genomic RNA modification profiles and that those differences appear to influence the translation of the viral RNA and viral growth in HEK-293 cells [18].

When infecting vertebrate hosts, arboviruses must overcome the innate immune response that is triggered in defense. Alphaviruses specifically counteract this response in a few ways; by shutting off host transcription and translation separately as well as by specifically targeting steps in the activation of type-I interferons (IFNs) [21,22,23,24]. Host transcriptional shutoff by alphaviruses is caused by the nonstructural protein 2, nsP2. NsP2 is translocated to the nucleus where it shuts off host transcription by degrading the RNA polymerase subunit RPB1 [22,23,24]. Some alphaviruses have been shown to shut off host translation through PKR activation and phosphorylation of eukaryotic initiation factor 2 (eIF2α), though there is evidence of additional mechanisms by which alphaviruses shut-off translation such as the recent identification of host translational shut-off being driven by phosphorylation of eukaryotic translation elongation factor 2 (eEF2) [22,23,25,26]. Finally, in addition to host transcriptional and translational shutoff, alphaviruses have been shown to directly antagonize type I IFN response by preventing the phosphorylation and nuclear translocation of signal transducers and activators of transcription 1 and 2 (STAT1 and STAT2) [21].

In this study, we investigate a human epithelial cell line that is capable of innate immune signaling in response to infection with mosquito- or mammalian-derived SINV. We hypothesize that the establishment of infection in epithelial cells may also play a role in the dissemination of alphavirus in a mammalian host alongside dendritic cell infection. Here, we performed the RNA sequencing analysis of Human Embryonic Kidney cells (HEK-293s) to elicit the differences in response to alphavirus infection derived from either vertebrate (Baby Hamster Kidney-21, BHK-21) cells or mosquito (Aedes albopictus C7/10) cells. We show that the mosquito-derived virus significantly alters the host response compared to the mammalian-derived virus. Interestingly, the mosquito-derived virus also induces more IFN-β and ISG mRNAs than the mammalian-derived virus. However, we show that while there are more IFN- β and ISG transcripts produced in the mosquito-derived virus infection, this is counteracted by rapid translational shutoff.

## 2. Materials and Methods

### 2.1. Insect and Mammalian Cell Culture

C7/10 Aedes albopictus cells were grown at 28 °C under 5% ambient CO_2_ in 1× Minimal Essential Media (Corning, Corning, NY, USA) supplemented with 10% heat-inactivated fetal bovine serum (Corning, Corning, NY, USA), 1% L-glutamine (Corning, Corning, NY, USA), 1% non-essential amino acids (Corning, Corning, NY, USA), and 1% antibiotic–antimycotic solution (Corning, Corning, NY, USA). Vertebrate baby hamster kidney fibroblast or BHK-21 cells as well as human embryonic kidney epithelial or HEK-293 cells were grown at 37 °C under 5% ambient CO_2_ in the same 1× Minimal Essential Media (Corning, Corning, NY, USA) as the C7/10 cells.

### 2.2. Virus Generation and Growth

BHK-21-derived viruses were obtained by in vitro transcription of pToto1101 SINV full-length plasmid using SP6 RNA polymerase (NEB, Ipswich, MA, USA). IVTs were then transfected into confluent BHK cells using the LTX transfection mix (Invitrogen, Waltham, MA, USA), following the manufacturer’s protocol, in a T25 flask (Greiner Bio-One CellStar, Monroe, NC, USA) with serum-free virus production media (SF-VPM) (Gibco, Waltham, MA, USA). Four hours after transfection, the SF-VPM was removed and replaced with fresh 1× MEM supplemented with 10% FBS. A total of 24 h after transfection, the viral supernatant was harvested and purified by centrifugation at 43,000× *g* for 2.5 h over a 27% *w*/*v* sucrose cushion in an HNE buffer (150 mM NaCl, 20 mM HEPES, 0.1 mM EDTA). The media was discarded, and the viral pellet was resuspended in 500 µL of HNE buffer. Plaque assays on BHK-21 cells were then performed to determine viral titer.

The C7/10-derived virus was obtained by infecting confluent C7/10 cells in a 150 mm dish (CellStar) with a P0 BHK-derived virus at an MOI of 0.1 PFU/cell. The inoculum was left on for 3 h, after which the media was replaced, cells were washed with 1× PBS (Phosphate Buffered Saline), and fresh 1× MEM was added. A total of 48 h after infection, the viral supernatant was harvested, purified and plaque assayed as previously described. All viruses were titered on BHK-21 cells and that titer was used for assay on HEK-293 cells.

### 2.3. RNA Sequencing

HEK-293 cells were infected in a 24-well plate with SINV at an MOI of 5 PFU/cell (three independent biological replicates per experimental condition). Cells were harvested at 8 h after infection and RNA was isolated using the BioRad (Hercules, CA, USA) Aurum Total RNA mini kit. Equimolar amounts of total RNA were submitted to Indiana University’s Center for Genomics and Bioinformatics for cDNA library construction using a TruSeq Stranded mRNA LT Sample Prep Kit (Illumina, San Diego, CA, USA) following the standard manufacturing protocol. Sequencing was performed using an Illumina NextSeq500 platform with a 75 bp cycle module generating 42 bp paired-end reads. After the sequencing run, demultiplexing was performed with bcl2fastq v2.20.0.422. NextSeq reads were trimmed using fastp (version 0.20.1) with parameters “-l17--detect_adapter_for_pe-g-p” [27]. The resulting reads were mapped against GRCh38 using HISAT2 (version 2.2.1) with parameters “--rna-strandness F” [28]. HISAT uses Bowtie2, which is based on the Burrows–Wheeler transformation algorithm, for sequence alignment and allows mapping across exon junctions [29]. Read counts for each gene were created using featureCounts from the Subread package (version 1.6.4) with the parameters “-O-M--primary-p--largestOverlap-s2-B” and Gencode v37 as the annotation [30,31]. Differential expression analysis was performed using the DESeq2 package (version 1.30.1) in R/Bioconductor (R version 4.0.4) [32]. Gene Ontology (GO) terms were assigned to genes using the Bioconductor package biomaRt (version 2.46.3) [33,34]. Cytoscape v3.9.1 was used to visualize genes corresponding to the GO term Type I Mediated Interferon Response and color was assigned to each gene based on the difference of log2 fold change between the mosquito-derived virus infection and the mammalian-derived virus infection. Full RNAseq data set was deposited in NCBI GEO. Accession numbers are GSE234344, and GSM7465398 through GSM7465406.

### 2.4. Real-Time Quantitative RT-PCR and Relative Expression Analyses

HEK-293 cells were infected in a 24-well plate with SINV at an MOI of 5 PFU/cell or mock infected. At 8 h after infection, the HEK-293 cells were resuspended in the media and spun down at 8000× *g* for 5 min. Cellular pellets were then RNA extracted using the BioRad Aurum Total RNA mini kit according to the manufacturer’s protocol. The extracted cellular RNA was then used as a template to synthesize cDNA using MMuLV Reverse Transcriptase (NEB) with random hexamer primers (Integrated DNA Technologies, Coralville, IA, USA) or primers specific to SINV E1. Quantitative RT-PCR analyses were performed using the Brilliant III SYBR green QPCR master mix (Thomas Scientific, Swedesboro, NJ, USA) with gene-specific primers according to the manufacturer’s protocol and with the Applied Bioscience StepOnePlus qRT-PCR machine (Life Technologies, Carlsbad, CA, USA). The expression levels were normalized to the endogenous GAPDH expression using the delta–delta comparative threshold method (ΔΔCT). Relative fold changes were determined using the comparative threshold cycle (CT) method compared back to the mock-infected cells for each gene assayed. For quantification of viral genomes, equal weight of RNA was added to each RT reaction using SINV E1-specific primers, and a standard curve comprising linearized SINV infectious clone containing the full-length genome. A minimum of three independent biological replicates were analyzed. Each biological replicate was measured as described above three times (technical replicates), and the average of these measurements for each biological replicate was used in the analyses shown.

### 2.5. IFN Bioassay

Type I interferon levels in cell culture supernatants were measured by interferon bioassay as described previously [35,36,37]. HEK-293 cells were seeded in a 24-well plate and infected with mosquito- or mammalian-derived SINV at an MOI of 5 PFU/cell. The entire supernatant was removed at each time indicated. All supernatant samples were acidified to a pH of 2.0 for 24 h and then neutralized to pH 7.4. The samples were then further inactivated by UV light for 10 min. Treated supernatant was then added to naïve HEK-293 cells seeded in a 96-well plate and titrated by twofold dilutions down the plate. Twenty-four hours after the addition of the treated supernatant, the interferon-sensitive SINV with an nsP3 fused GFP was added to each plate at an MOI of 10 PFU/cell. Twenty-four hours after the addition of the virus, the infected cells were measured by green object count using an IncuCyte live-cell analyses system (Essen Biosciences, Ann Arbor, MI, USA). An IFN-β standard was added to each plate to determine inhibitory units/mL based on green object count (virus-positive cells). Three independent biological replicates for each virus infection were analyzed.

### 2.6. Western Blotting

HEK-293 cells were synchronously infected at 4 °C for 30 min with C7/10- or BHK-derived SINV at an MOI of 5 PFU/cell. At the indicated times after infection, HEK-293 cells were treated with whole cell lysis buffer (1% NP-40, 10 mM Tris pH 7.4, 20 mM NaCl, 1 mM EDTA, 1× SDS buffer) before being boiled at 95 °C for 30 min. Equal volumes of whole cell lysates were analyzed via 10% SDS-PAGE using the antibodies indicated. The gel was transferred to a PVDF membrane and blocked in a 5% TBSM (Tris Buffered Saline 1× + 5% Dry Milk) for 1 h at room temperature. Following blocking, the membrane was incubated with Anti-IFIT1 (#14769), anti-beta-actin (#4967), anti-eEF2 (#2332), or anti-phospho-eEF2 (# 2331) all Cell Signaling Technology at 4 °C overnight. Following incubation with the primary antibody, the membrane was washed with the TBST (Tris Buffered Saline 1× + 0.1% Tween-20) prior to incubation with goat anti-rabbit AlexaFluor 750 (#A-21039 ThermoFisher, Waltham, MA, USA) or anti-rabbit IgG, HRP-linked antibody (#7074 Cell Signaling, Danvers, MA, USA) for 1 h at room temperature. Following secondary antibody incubation, the membrane was imaged with radiography film (for HRP-linked antibody) or using a BioRad ChemiDoc MP Imaging System (for AlexaFluor antibody). Band intensity was determined using ImageStudio Lite Ver. 5.2. Band intensity is represented as a ratio of IFIT1 over actin standardized to the BHK-derived virus signal or as a ratio of eEF2 or phospho-eEF2 over actin and finally as a ratio of phospho-eEF2/actin over eEF2/actin. For each protein, a minimum of three biological replicates were analyzed.

### 2.7. Translational Activity Measured by [^35^S] Methionine Radiolabeling

HEK-293 cells were synchronously infected for 30 min at 4 °C with either BHK-21 or C7/10-derived SINV at an MOI of 5 PFU/cell in a 24-well plate. A total of 30 min prior to the labeling period, the infected cells were washed with 1× PBS and incubated with methionine/cysteine-free DMEM. After the depletion period, the media were removed and replaced with fresh methionine/cysteine-free DMEM supplemented with [^35^S]-labeled methionine and cysteine (Express Protein Labeling Mix, Perkin Elmer, Waltham, MA, USA) at a specific activity of 50 µCi/mL and incubated for one additional hour. Cells were treated with a cytoplasmic lysis buffer (1% Triton-X100, 10 mM Tris pH 7.4, 20 mM NaCl, 1 mM EDTA, 1× PMSF), then incubated on ice for 10 min while vortexing intermittently every 2 min. Samples were then spun down for 10 min at 10,000× *g* before the supernatant was removed and placed in a new tube. A 6× SDS running buffer was then added to the cytoplasmic lysates to a concentration of 1× and the lysates were boiled for 5 min. Equal volumes of cytoplasmic lysates were analyzed via 10% SDS-PAGE, and radiolabeled proteins were detected by autoradiography. Actin was used as a proxy for host translation. Band and lane intensities were determined using ImageStudio Lite Ver. 5.2. All comparisons in this manuscript are made only within the lanes from the same gel on a single autoradiographic exposure. Three independent infections per virus sample were performed and assayed as described (*n* = 3) to generate the data shown.

### 2.8. Statistical Analysis of Experimental Data

All statistical analyses were performed using GraphPad Prism 9 (GraphPad Software Inc., San Diego, CA, USA).

## 3. Results

### 3.1. Host Derivation of Virus Influences Type I IFN Mediated Transcriptional Response

We aimed to determine the role that the host cell response was playing in the previously observed differences in replication kinetics between the mosquito- and mammalian-derived virus [18,38]. Previous studies, although in different cell lines and a different alphavirus, have suggested that mosquito-derived alphaviruses induce less IFN-β compared to their mammalian-derived counterpart [35]. In a previous study, we examined the growth kinetics of mosquito- and mammalian-derived viruses in HEK-293 cells and observed that while early growth kinetics were similar between the viruses of different origin, the mosquito-derived virus displayed a growth advantage beginning at approximately 8 h after infection [18]. Using the same viruses in this study, we asked whether the mammalian derived virus was inducing a more robust innate immune response causing it to replicate more poorly when compared to its mosquito-derived counterpart. RNA sequencing analyses comparing transcriptional responses of cells infected with mosquito- or mammalian-derived alphaviruses have not been previously performed. We examined the transcriptional response of HEK-293 cells infected with BHK-derived or C7/10-derived SINV at an MOI of 5 PFU at 8 h after infection as described in the Materials and Methods section. DeSeq2 was performed on RNA-seq data with a 0.05 adjusted *p*-value cutoff for significance. Genes with an adjusted *p*-value of greater than 0.05 were excluded from analysis. The RNA-seq data showed a greater overall response to the C7/10-derived virus than to the mammalian-derived virus by HEK-293 cells (Figure 1A,B). There were 1932 protein coding genes differentially expressed with a log2 fold change of greater than 1 or less than −1 for infection with either virus versus the mock-infected control. We compared the degree of expression change between the C7/10 virus infection and the BHK virus infection. As can be seen from the scatter plot in Figure 1B, infection with the C7/10-derived virus lead to a greater degree of expression change for the majority of these genes, as shown by them falling above the line designating a linear relationship.

On the basis of previous studies that showed a difference in type I interferon response to virus derived from different hosts [35], we examined all genes from the GO term “type I interferon-mediated signaling pathway” and visualized log2 fold change differences between the mosquito-derived infection and the mammalian-derived infection using a Cytoscape (Figure 1C). A heat map was generated, taking the difference in log2 fold change in the mammalian-derived infection from the log2 fold change in mosquito-derived infection (Figure 1C). Surprisingly, we found that while both viruses are inducing IFN-β transcripts, the mosquito-derived virus is inducing it by 4.4 log2-fold change greater than the mammalian-derived virus (Figure 1C). Also surprising, though not unexpected given the levels of IFN-β transcripts, is the greater induction of multiple ISG transcripts (Figure 1C). The interferon-induced protein with tetratricopeptide repeats 1 (IFIT1), for example, is being induced at a 2.8 log2-fold change increase in the mosquito-derived virus infection compared to the mammalian-derived virus infection (Figure 1C). In addition to RNA-seq, qRT-PCR was performed on HEK-293 cells infected with C7/10- or BHK-derived virus at an MOI of 5 PFU at 8 h after infection. The qRT-PCR results confirm the RNA-seq analysis and show a greater induction of IFN-β as well as ISG transcripts (Figure 2). More specifically, there was approximately a 26.2-fold increase of IFN-β transcripts present in the C7/10-derived virus infection compared to the BHK-derived virus infection. This trend follows for the other ISG transcripts analyzed, with ISG15 having an approximately 7.1-fold increase, IFIT1 an 11.7-fold increase, and OASL a 13.2-fold increase in transcripts in the C7/10-derived virus infection compared to the BHK-derived virus infection. Unlike in the immune cells previously examined, specifically myeloid dendritic cells, the mosquito-derived virus is inducing more IFN-β transcripts in epithelial cells [35]. It is worth noting, however, that the previous study was conducted at different time points, in a different cell type, and with a different alphavirus [35]. While the C7/10-derived virus replicates better in mammalian cells that have (HEK-293) or lack (BHK-21) an intact innate immune response suggesting the virus is “primed” for infection of mammalian cells, the induction of IFN-β and ISG transcripts seems to conflict with the enhanced replication [15].

### 3.2. Translation of ISGs Contrasts with Type I IFN-Mediated Transcriptional Response

Having observed a stronger transcriptional response in the HEK-293s infected with the mosquito-derived virus, we performed an inhibition assay to determine the levels of type I IFN being expressed by the cells. Briefly, HEK-293 cells were infected with C7/10- or BHK-derived SINV at an MOI of 5. At the times indicated, the supernatant was harvested, acidified and UV-treated to remove any virus present, then added to naïve HEK-293 cells before being infected with a fluorescently tagged SINV. Inhibition was determined by infecting the pre-treated cells with SINV at an MOI of 10 and measuring the number of infected cells via green fluorescence as measured by Incucyte. Using this method, we observed a significantly increased amount of type I IFN being produced in the mosquito-derived virus-infected cells at 4 hpi, approximately 1.5-fold more (Figure 3A). When using this same method but now looking at 24 hpi, we observed a significant increase in the amount of type I IFN in the BHK-derived virus-infected cells, approximately a 3.5-fold increase (Figure 3A). This later time point appears to be more in line with the previously reported findings [35]. Also of interest is the change in the amount of type I IFN produced between 4 and 24 hpi in the C7/10- or BHK-derived virus infections. The C7/10-derived virus infection essentially produces the same amount of type I IFN as determined by our assay at both 4 and 24 hpi (Figure 3A). Meanwhile, the BHK-derived virus infection had an approximately 5-fold increase in type I IFN produced between 4 and 24 hpi (Figure 3A). Given the conflict between enhanced replication of the mosquito-derived virus and the transcriptional response in HEK-293 cells and compounded by the inhibition assay suggesting there is no increase in type I IFN production after 4 hpi, we hypothesized that host cell translation may be shut off more rapidly even though the transcription of innate response genes was high. Thus, we performed Western blots to measure the expression levels of IFIT1. Briefly, HEK-293 cells were infected with either mosquito- or mammalian-derived SINV at an MOI of 5 PFU and total cell extracts were harvested at 12 hpi. We found that the mosquito-derived virus has less IFIT1 protein expressed at 12 hpi, with there being approximately 3.2 times as much IFIT1 protein in the BHK-derived virus infection compared to the C7/10-derived virus infection (Figure 3B,C). We also found that IFIT1 mRNA is present at approximately 5.6-fold higher levels in the C7/10-derived virus infection than in the BHK-derived virus-infected cells (Figure 3D). Finally, to confirm previous findings and ensure that viral genome replication was not playing a role, we determined viral genome copies in the HEK-293 cells at 12 hpi (Figure 3E) [18]. We found that there is no significant difference in the amount of viral genome present in the HEK-293 cells at 12 hpi (Figure 3E). This result suggests that the mosquito-derived virus is inducing a stronger transcriptional response than the mammalian-derived virus, but that this greater transcriptional response is being counteracted by what we hypothesized to be a faster translational shutoff (Figure 3B–D).

### 3.3. Host Derivation of Infecting Virus Influences Speed of Translational Shut-Off

After observing the disconnect between IFIT1 transcript levels and protein levels in the C7/10-derived virus infections, we hypothesized that the C7/10-derived virus is shutting off host translation sooner and preventing the translation of these transcripts, thus explaining the disconnect between transcript levels and protein levels as well as how the C7/10-derived virus is able to replicate better in the presence of more IFN-β and ISG mRNAs. Translational and transcriptional shut-off occur at different times in infection for alphaviruses [26,39]. We examined translational shu-toff by pulse labeling of proteins at different times after infection. HEK-293 cells were infected with a C7/10- or BHK-derived virus at an MOI of 5 PFU. From 4 to 8 hpi, cells were labeled for 1 h with [^35^S] methionine and harvested. Proteins were separated by SDS-PAGE and visualized by autoradiography. We found that the C7/10-derived virus is shutting off host translation more rapidly than the BHK-derived virus (Figure 4A,B). The band on the gel that corresponds to β-actin (arrow) was used as a proxy for host translation at each time point. By 8 hpi, the actin band is barely visible for the C7/10-derived virus infection sample, whereas for the BHK-derived virus infection sample, actin is still being translated (although reduced compared to earlier timepoints). Quantifying the levels of actin as a proportion of total protein translated at 8 hpi, we see an approximately 3.4-fold greater amount of protein represented as actin in the BHK-derived virus infection compared to the C7/10-derived virus infection (Figure 4B). Viral capsid protein is also indicated with a band, showing corresponding viral replication happening for both mammalian- and mosquito-derived virus infections (Figure 4A).

### 3.4. Mosquito-Derived Virus Infection Leads to Elevated Levels of Phosphorylated eEF2 Compared to Mammalian-Derived Virus Infection

To confirm that mosquito-derived SINV is indeed shutting off host translation sooner, we performed Western blots to measure the level of phosphorylated eukaryotic translation elongation factor 2 (eEF2). It has been recently reported that phosphorylation of eEF2 is at least partially responsible for the SINV-induced host translational shut-off [26]. Thus, we examined eEF2 phosphorylation levels in cells infected with the C7/10-derived virus and the BHK-derived virus. We found that at 10 hpi, the C7/10-derived virus infection led to a significantly higher proportion of phosphorylated eEF2 compared to total eEF2 (Figure 5A,C). There was no significant difference in the amount of eEF2 protein present, but there was an approximately 2.5-fold increase in the amount of phosphorylated eEF2 in the C7/10-derived infection (Figure 5B,C). To add to this, when comparing the relative amount of phosphorylated eEF2 to that of eEF2, we found a 2.6-fold increase in the ratio in C7/10-derived virus infection compared to that of the BHK-derived virus infection (Figure 5D). These data suggest that the C7/10-derived virus is leading to increased phosphorylation of eEF2 in the HEK-293 cells resulting in more rapid host translational shut-off.

## 4. Discussion

Arboviruses are defined by their capacity to replicate and infect both vertebrate and arthropod hosts. Understanding the differences in insect- and vertebrate-derived arboviruses and how those differences determine outcome of infection may inform the development of transmission interventions. To our knowledge, there has not been an in-depth look at the differences in response in a mammalian epithelial cell infected with alphaviruses derived from mosquito or mammalian cells. Here, we show that in HEK-293 cells, mosquito-derived SINV elicits a more robust transcriptional response than the BHK-21-derived counterpart, yet it counters the innate immune response through translational inhibition.

Along with other known differences in host-acquired modifications of arboviruses, such as differential glycosylation or packaging of host molecules, we showed that differential modifications of the viral RNA genome also influence the outcome of infection for SINV [13,14,18]. Cell type plays a strong role in the outcome and response to arbovirus infection, and while studies have been conducted that show a similar, though not identical, phenotype of mosquito- and mammalian-derived arboviruses, the differential response of these two viral origins has only been examined in immune cells [35]. We showed a response in epithelial cells capable of innate immune signaling. A different response has been observed in immune cells, and this may reveal more about the timing of the infection [35]. Mosquito-derived virus on myeloid dendritic cells (mDCs) has been shown to replicate better than its mammalian-derived counterpart, similar to our observations in epithelial cells, though the difference we observed in the transcriptional response of IFN-β is starkly contrasting [35]. It is important to point out, however, that our observed type I IFN production from the IFN bioassay at 24 hpi very closely resembles what was previously shown (Figure 3A) [35]. Meanwhile, we observed surprising differences at earlier timepoints in infection (Figure 1, Figure 2 and Figure 3A). These results may shed light on the complexity of studying the differences in mosquito- and mammalian-derived arboviruses and clarify the importance of timing for future studies. We show that while the mosquito-derived virus is causing a drastic increase in the amount of IFN-β and ISG mRNA, it is counteracting this increase in transcripts by shutting off host translation sooner. This more rapid translational shut-off can be, at least partially, attributed to the increased amount of phosphorylated eEF2 present in the cells infected with the mosquito-derived virus. Future studies will look to determine a more detailed mechanism of the ways in which the mosquito-derived virus is shutting off host translation sooner and could explore the role of the differential RNA modification profile’s impact on this observed phenotype. From these results, we hypothesize that the mosquito-derived virus enhances replication kinetics in the mammalian host, which leads to increased PAMPs that are detected in the early stages of infection, resulting in the enhanced type I mediated IFN transcriptional response at 4 and 8 hpi (Figure 1, Figure 2 and Figure 3). However, this enhanced transcriptional response appears to be quickly counteracted by the more rapid shut-off of host translation by the mosquito-derived virus, as we observed reduced type I IFN produced at 24 hpi, as well as reduced IFIT1 protein expression at 12 hpi, and finally a greater accumulation of phosphorylated eEF2 at 10 hpi (Figure 3A, Figure 4 and Figure 5).

This altered host response to infection by mosquito- or mammalian-derived virus may shed light on the complex initial bite from an infected mosquito. Infection of mDC’s and other local skin immune cells are known to be an important step in dissemination of the virus throughout the mammalian host [4,5,6,7]. These cells may be directly infected by the incoming mosquito-derived virus or by the virus produced in epithelial cells infected following the mosquito bite. Recruitment of epidermal immune cells to the site of the bite and hence virus deposition is enhanced by the inflammatory response evoked through the bite and the components of the mosquito saliva [3,4,7,8,9]. Therefore, we hypothesize that turning off the innate immune response enhances virus replication in the initially infected epithelial cells, increasing virus presence in the area of the bite and consequently increasing the chances of infecting mDCs and dissemination.

Mosquito- and mammalian-derived alphaviruses differ in their efficiency of infection in mammalian cells, suggesting they have been “primed” for future infection by the host in which they were produced in. We know that the particles produced in mosquito cells differ from those produced in mammalian cells. The glycosylation of the surface glycoproteins of the mosquito-produced virus leads to increased binding and infectivity of cells expressing C-type lectins, such as dendritic cells [14]. There is also evidence in the Ross River Virus that N-linked glycans were essential for induction of IFN by the mammalian-derived virus [39]. However, it seems unlikely that this acquisition is playing a role in the phenotypes we report here and previously [18]. We also know that a sub-population of the mammalian-derived virus and the mosquito-derived virus contain host ribosomal components which could be influencing the replication differences between the viruses from different hosts and/or the host response to infection by either virus [15]. It is possible that these host ribosomal components are playing a role in host response; however, we have previously reported that purified vRNA obtained from mosquito-derived virions has increased translational activity compared to vRNA obtained from mammalian-derived virions [18]. It is important to note, however, that while the translation of the vRNA appears to be enhanced independent of host ribosomal components, it is possible that these ribosomal components are influencing the host response to infection with virions derived from mosquito or mammalian cells [18]. Another possibility is that the viruses from different hosts encapsidate vRNA that differ in structure such as defective genomes. Additionally, the proportion of capped genomic RNA has been shown to differ between mosquito-derived and mammalian-derived SINV [40]. Finally, we have recently shown that the RNA modification profile of the genomic vRNA derived from mosquito and mammalian cells is significantly different, indicating that SINV acquires host-specific modifications of its genomic RNA [18]. N6-methyladenosine (m6A) has previously been shown to influence flavivirus replication in mammalian cells, and 5-methylcytosine (m5C) modifications have been shown to influence alphavirus replication in arthropod cells [19,20]. RNA modifications have been shown to reduce innate immune signaling by disrupting RIG-I detection of RNA [41]. For instance, m6A modifications specifically have been shown to enable human metapneumovirus to escape recognition by RIG-I [42]. It is therefore possible that the differences in RNA modification profile we reported between mosquito- and mammalian-derived SINV is also influencing the transcriptional response to infection that we show here [18]. The main molecular consequences of infection with the mosquito-derived virus we observed are the increased translation of viral RNA [18] and a more rapid shut-off of host cell translation. In the future, it will be interesting to determine whether and in what ways the RNA modification profile of infecting viral RNA impacts these processes directly.

## Figures and Tables

**Figure 1 viruses-15-01685-f001:**
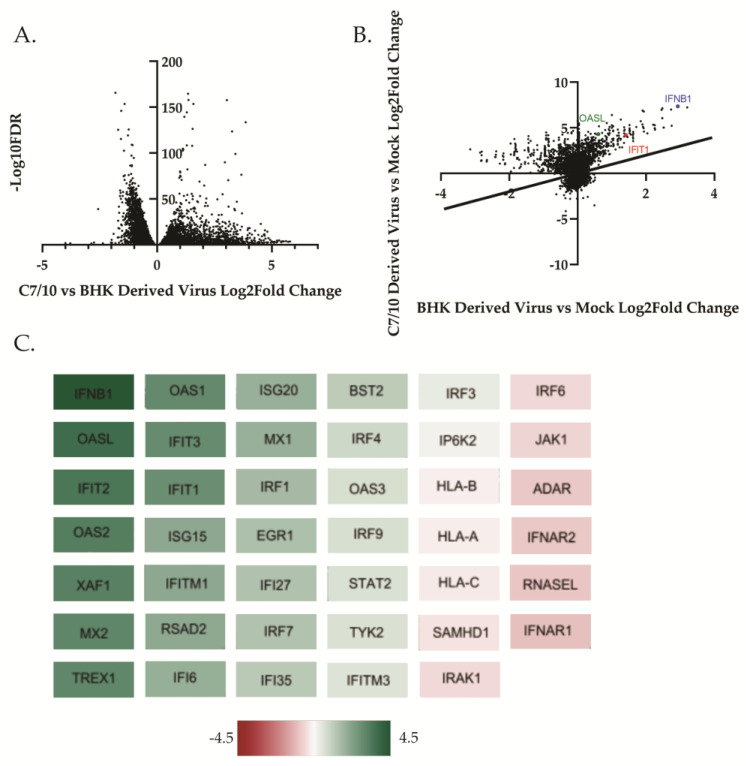
Host derivation of virus influences transcriptional response in HEK-293 cells. Principal component analysis of RNA-Seq data set was generated using DESeq2 as described in Materials and Methods. (**A**) Volcano plot of the difference in log2 fold change in each gene differentially expressed following infection with either C7/10- or BHK-derived SINV compared to mock-infected cells is graphed with a cutoff of padj < 0.05. (**B**) Scatter plot of log2 fold change in C7/10-derived virus infection versus BHK-derived virus infection. C7/10-derived virus versus mock log2 fold change is graphed on the *y*-axis against log2 fold change in BHK-derived virus infection compared to mock on the *x*-axis. The IFNB1 (blue) gene as well as IFIT1 (red) and OASL (green) are highlighted. Cutoff was set at a padj < 0.05 when comparing the difference of BHK-derived DEGs from C7/10 DEGs. The black line represents the linear relationship of DEGs between the two virus infections such that DEGs that fall on the line are equally differentially expressed in both virus infections. (**C**) Differentially expressed genes from the RNA-Seq data which correspond to the Gene Ontology (GO) term “Regulation of Type I Interferon Mediated Signaling Pathway” were visualized using Cytoscape. Scale bar represents the difference in log2 fold change of C7/10-derived virus infection compared to BHK-derived virus infection.

**Figure 2 viruses-15-01685-f002:**
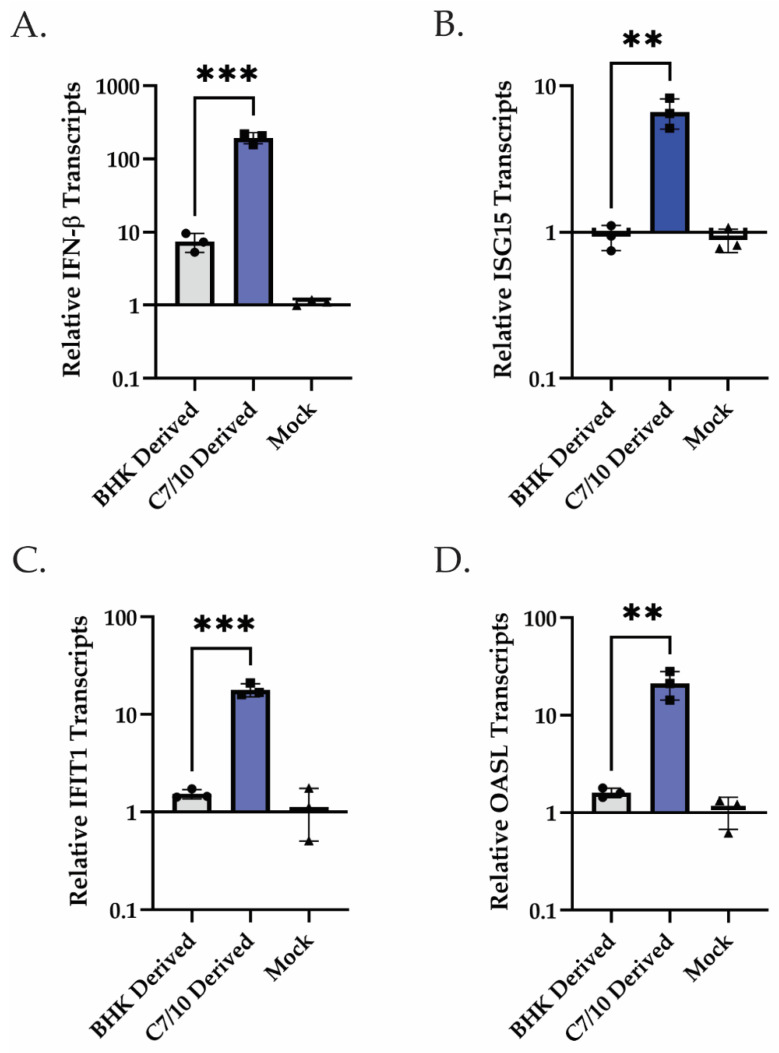
C7/10-derived virus infection leads to elevated levels of IFN-β, ISG15, IFIT1, and OASL transcripts compared to BHK-derived virus infection in HEK-293 cells. qRT-PCR analysis of IFN-β and ISGs performed on extracted RNA harvested at 8 hpi following infection with either C7/10- or BHK-derived SINV. (**A**) qRT-PCR analysis of IFN-β transcripts at 8 hpi following infection with C7/10- or BHK-derived virus represented as fold change relative to those of mock-infected samples. Error bars represent SEM of biological replicates (*n* = 3). Student’s *t*-test: *p* *** = 0.0006. (**B**) qRT-PCR analysis of ISG15 transcripts at 8 hpi following infection with C7/10- or BHK-derived virus represented as fold change relative to those of mock infected samples. Error bars represent SEM of biological replicates (*n* = 3). Student’s *t*-test: *p* ** = 0.0032. (**C**) qRT-PCR analysis of IFIT1 transcripts at 8 hpi following infection with C7/10- or BHK-derived virus represented as fold change relative to those of mock-infected samples. Error bars represent SEM of biological replicates (*n* = 3). Student’s *t*-test: *p* *** = 0.0005. (**D**) qRT-PCR analysis of OASL transcripts at 8 hpi following infection with C7/10- or BHK-derived virus represented as fold change relative to those of mock-infected samples. Error bars represent SEM of biological replicates (*n* = 3). Student’s *t*-test: *p* ** = 0.0078.

**Figure 3 viruses-15-01685-f003:**
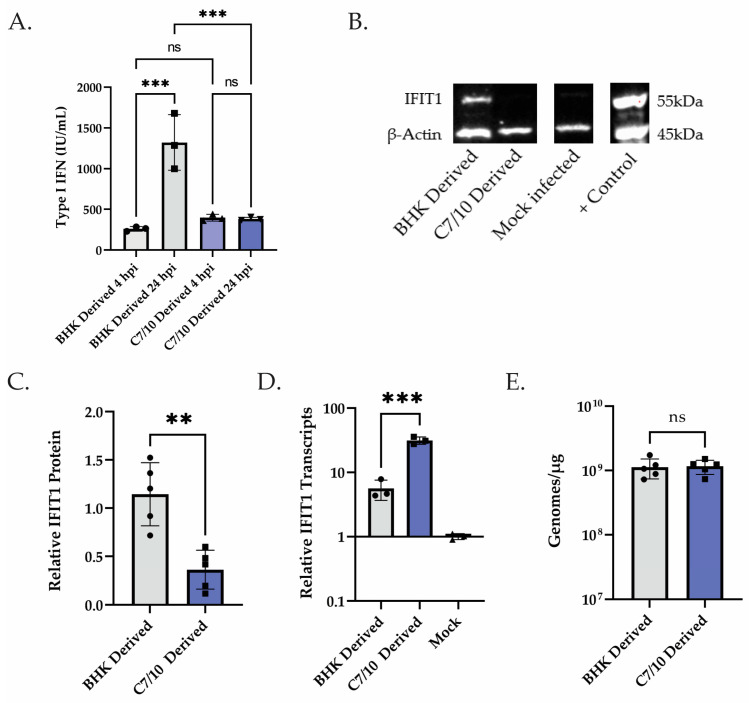
Translation of ISGs contrasts with type I IFN-mediated transcriptional response. (**A**) Supernatant obtained at 4 and 24 hpi from HEK-293 cells infected with either C7/10- or BHK-derived virus was used to pre-treat naïve HEK-293 cells to measure type I IFN produced via IFN Bioassay as described in Materials and Methods. Error bars represent SEM of biological replicates (*n* = 3). One-way ANOVA with multiple comparisons: ns = not statistically significant. BHK 4 hpi to BHK 24 hpi *p* *** = 0.0003. BHK 24 hpi to C7/10 24 hpi *p* *** = 0.0007. (**B**) Western blot analysis of HEK-293 cells infected with BHK- or C7/10-derived virus, control lane was transfected with dsRNA from UV-inactivated reovirus. Samples collected at 12 hpi and probed for IFIT1 and β-actin. (**C**) Quantified levels of IFIT1/actin band intensity from (**C**) relative to the BHK-derived virus infection. Error bars represent SEM of biological replicates (*n* = 5). Student’s *t*-test: *p* ** = 0.0019. (**D**) qRT-PCR analysis of samples from (**C**) to determine levels of IFIT1 transcripts using GAPDH as a control and set relative to mock-infected HEK-293s. Error bars represent SEM of biological replicates (*n* = 3). Student’s *t*-test: *p* *** = 0.0006. (**E**) qRT-PCR analysis of samples from (**C**) to determine viral genome replication in the cells. Genomes per microgram of total cellular RNA determined using a standard curve comprising linearized SINV infectious clone containing the full-length genome. Error bars represent SEM of biological replicates (*n* = 5). Student’s *t*-test: ns = not statistically significant.

**Figure 4 viruses-15-01685-f004:**
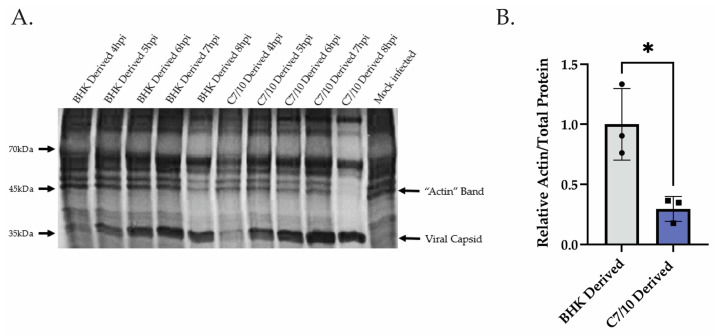
C7/10-derived virus shuts off host translation sooner than mammalian-derived virus. (**A**) Translational activity measured by pulse-chase [^35^S] methionine radiolabeling. Each time point is representative of protein translated within the hour leading up to the time point listed. The band corresponding to β-actin is labeled with an arrow, as well as the band corresponding to viral capsid (**B**) Quantitative measurement of the amount of actin at 8 hpi compared to the total amount of protein within the same lane. The actin/total protein ratio is standardized to the BHK-derived virus infection. Error bars represent SEM of biological replicates (*n* = 3). Student’s *t*-test: *p* * = 0.0181.

**Figure 5 viruses-15-01685-f005:**
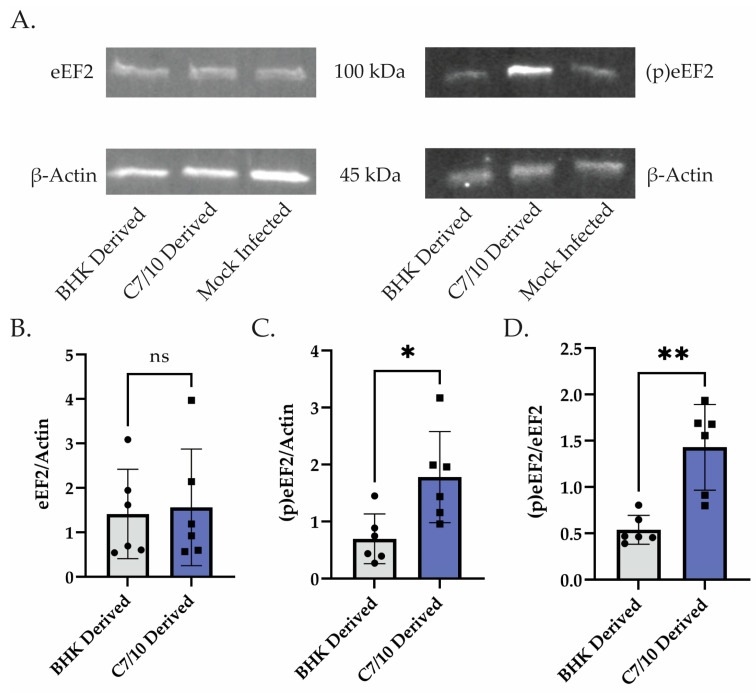
Mosquito-derived virus infection leads to elevated levels of phosphorylated eEF2. (**A**) Western blot analysis of HEK-293s infected with C7/10- or BHK-derived virus and harvested at 10 hpi. Left is blotted with eEF2 and β-actin antibody. Right is probed with phosphorylated eEF2 and β-actin antibody. (**B**) Quantitative analysis of the band intensity of eEF2 represented as a ratio to actin within the same lane. Error bars represent SEM of biological replicates (*n* = 6). Student’s *t*-test: ns = not statistically significant. (**C**) Quantitative analysis of the band intensity of phosphorylated eEF2 represented as a ratio to actin within the same lane. Error bars represent SEM of biological replicates (*n* = 6). Student’s *t*-test: *p* * = 0.0155 (**D**) Ratio of the analyses performed in (**B**,**C**) where a single point represents the ratio of phosphorylated eEF2/actin compared to eEF2/actin for the same biological replicate. Error bars represent SEM of biological replicates (*n* = 6). Student’s *t*-test: *p* ** = 0.0012.

## Data Availability

The complete RNAseq data set has been deposited in NCBI GEO Accession numbers GSE234344, and GSM7465398 through GSM7465406.

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
