# Peer review of "Host Derivation of Sindbis Virus Influences Mammalian Type I Interferon Response to Infection"

_viruses, 2023, doi:10.3390/v15081685_

Round 1

Reviewer 1 Report

Previously, the authors and others have shown that mosquito derived virus can more efficiently infect mammalian cells than mammalian derived virus, suggesting host-specific adaptations. In this study, the authors performed RNA-seq analysis to determine host responses induced by different host derived Sindbis viruses. They found that infection with mosquito derived viruses leads to greater transcriptional changes, especially in the IFN and ISG transcripts. Surprisingly, translation of ISGs is lower in 293 cells infected with mosquito derived viruses, consistent with faster host translational shutoff that is dependent on higher levels of eEF2 phosphorylation. This study addresses an important and interesting question as to how replication in alternating hosts might modulate the host response/environment in the mammalian host. However, the model of mosquito derived viruses causing enhanced transcriptional changes with faster translational shutoff needs to be strengthened.

Major comments:

1.     It is confusing to refer to 293 cells as “an immune competent human epithelial cell line” throughout the manuscript such as in Lines 121 and 474 when they are human embryonic kidney cells with epithelial morphology.

2.     In Figure 1C, most of the ISGs that were differentially regulated are more upregulated in mosquito derived virus infected 293 cells but some critical for IFN signaling are more downregulated including IFNAR1, IFNAR2, and JAK1, which can negatively impact the IFN pathway. It would be informative to also show validation of some of these more downregulated genes, and to discuss how their downregulation might affect the overall antiviral state during mosquito vs mammalian derived virus infections.

3.     In Figure 3A-B, it would be helpful to combine these two panels into one graph with the same y-axis so it clearly shows that IFN levels are similar for cells treated with supernatant harvested from mosquito derived virus infected cells at 4 and 24 hpi.

4.     It is surprising that by 8 hpi there is no decrease in normalized actin protein levels in cells infected with a high MOI of mammalian derived viruses in Figure 4A-B.

5.     The authors should discuss why the band for phosphorylated eEF2 is larger in mock infected lysates in Figure 5A.

6.     The authors should provide more discussion on the factors that contribute to differences in induction of IFNB/ISG transcripts by infection with mosquito derived viruses in different cell types shown in current and previous studies.

7.     It is not clear how many independent experiments were performed for each figure.

Minor comments:

1.     Line 36: Chikungunya virus should be changed to chikungunya virus.

2.     Line 94: The sentence should be changed to “…viral kinetics in mammalian cells…”.

3.     Line 99: Flavivirus should be changed to flavivirus.

The introduction and conclusion sections need to be shortened and modified to provide more focus on the current study and findings.

Round 2

Reviewer 1 Report

The authors have not addressed the following comments:

1.     Regarding Figure 1C, it would be informative to show validation of some of the downregulated genes in the cover letter even if the authors did not find statistically significance differences. In addition, the manuscript would have benefited from a more thorough discussion of how the gene expression changes (both up- and down-regulation of IFN/ISGs) might have affected the overall antiviral state.

2.     It is still not clear whether at least two independent experiments were performed for each panel of the figures. Without that confirmation from the authors, the rigor of the study and significance of the findings are in question. For example, the authors proposed that faster translational shutoff by mosquito derived viruses is due to phosphorylation of eEF2 in Figure 5. It would be important to show that increased phosphorylated eEF2 is consistent across multiple experiments. Also, the authors stated that two independent experiments each with three biological replicates were performed to measure translational activity. Is Figure 4 representative of the two experiments?

3.     Line numbers should be clearly stated when the authors refer to the changes they have included in the revised manuscript.

The manuscript can be written in a more concise manner to avoid reading like a review article.

Reviewer 2 Report

In this revised paper by Crawford et al., the authors have responded appropriately to the concerns noted previously, including providing data obtained after infection of the cells with virus normalized by particle rather than plaque forming units. A few minor changes would further improve the manuscript.

1. The authors mention in their response letter that “It is possible that the genome/PFU ratio between the two virus stocks and/or presence of defective genomes is different, and this could lead to the different initial PAMP exposure. The determination of what such differences are between viruses of different host derivations is the focus of ongoing studies in the lab.” Perhaps this possibility would be worth mentioning in the discussion.

2. In the response letter they note that “the viruses used in the previously published manuscript for the growth curves in Fig 1 are the same preparations as used for the experiments in this manuscript under review. The mosquito derived virus does replicate faster in the sense of infectious virus released, however, at the time points we have looked at (8/12hpi) there is no apparent difference in total genomes present in the infected cell.” It would be helpful to include text in the results section 3.1 to emphasize that although SINV replication kinetics are different between the two virus stocks at later timepoints, that the early kinetics are similar at the timepoints studied in this current manuscript. Given the similar replication as determined by qPCR in this study or the prior study, the differences observed in innate immunity and host translation shut off are more convincing at these timepoints.
